# Large-scale network analysis captures biological features of bacterial plasmids

Mislav Acman [1✉], Lucy van Dorp [1], Joanne M. Santini [2] & Francois Balloux [1✉]

Many bacteria can exchange genetic material through horizontal gene transfer (HGT) mediated by plasmids and plasmid-borne transposable elements. Here, we study the population structure and dynamics of over 10,000 bacterial plasmids, by quantifying their genetic similarities and reconstructing a network based on their shared $k$-mer content. We use a community detection algorithm to assign plasmids into cliques, which correlate with plasmid gene content, bacterial host range, GC content, and existing classifications based on replicon and mobility (MOB) types. Further analysis of plasmid population structure allows us to uncover candidates for yet undescribed replicon genes, and to identify transposable elements as the main drivers of HGT at broad phylogenetic scales. Our work illustrates the potential of network-based analyses of the bacterial 'mobilome' and opens up the prospect of a natural, exhaustive classification framework for bacterial plasmids.

[1] UCL Genetics Institute, University College London, Gower Street, London WC1E 6BT, UK. [2] Institute of Structural and Molecular Biology, University College London, Gower Street, London WC1E 6BT, UK. ✉email: mislav.acman.17@ucl.ac.uk; f.balloux@ucl.ac.uk

Plasmids are extra-chromosomal DNA molecules found across all three Domains of Life. In bacteria, they are considered one of the main mediators of horizontal gene transfer (HGT) through the processes of conjugation and transformation[1–3]. Plasmids generally harbour non-essential genes that can modulate the fitness of their bacterial host. Some prominent examples include toxin–antitoxin systems, virulence factors, metabolic pathways, antibiotic biosynthesis, metal resistance and antimicrobial resistance (AMR) genes. These accessory genes can be located on transposable elements involved in lateral gene transfer across genomes and can thus lead to a highly mosaic structure of plasmid genomes[4]. The mix of vertical and horizontal inheritance of plasmids, together with exchanges of plasmid-borne genes, generates complex dynamics that are difficult to capture with classical population genetics tools and make it challenging to classify plasmids within a coherent universal framework.

Currently, there are two well-established plasmid classification schemes that attempt to bin plasmids according to their propagation mechanisms, while indirectly capturing some features of the plasmid backbone. The first scheme is based on replicon types[5] and the second on mobility (MOB) groups[6]. Replicon-based typing relies on relatively conserved genes of the replicon region, which encode the plasmid replication and partitioning machinery[5]. Plasmids with matching replication or partitioning systems cannot stably coexist within the same cell. Conversely, MOB typing is used to classify self-transmissible and mobilizable plasmids into six MOB types[6]. The MOB-typing scheme relies on the conserved N-terminal sequence of the relaxase, a site-specific DNA endonuclease that binds to the origin of transfer cleaving at the nic site and is essential for plasmid conjugation.

Despite being widely used and informative, these typing schemes only work within a limited taxonomic range[7–9]. Replicon typing is dependent on the availability of prior experimental evidence and remains restricted to culturable bacteria from the family *Enterobacteriaceae* and several well-studied genera of Gram-positive bacteria[1,10–12]. Furthermore, this approach can lead to ambiguous classification, even for experimentally validated replicons, as recently demonstrated by the discovery of compatible plasmids assigned to the same replicon type, which led to the further subdivision of the IncK type into IncK1 and IncK2[13], and IncA/C type into IncA and IncC[14]. In addition, plasmids can carry genes from more than one replication machinery, leading to assignment to multiple replicon types, further reducing interpretability[7,8]. MOB-typing schemes generate fewer multiple assignments and can cover a potentially wider taxonomic range; however, they are not applicable to the classification of non-mobilizable plasmids. These two typing schemes have inspired several in silico classification tools, such as PlasmidFinder[12], the plasmid MultiLocus Sequence Typing database and MOB suite[15]. However, all of these tools intrinsically rely on the completeness of their reference sequence databases, which typically lack representatives from understudied and/or unculturable bacterial hosts.

As bacterial plasmids undergo extensive recombination and HGT, their evolutionary history is not well captured by phylogenetic trees, which are designed for the analysis of point mutations in sequence alignments[16,17]. Network models offer an attractive alternative given they can incorporate both horizontal and vertical inheritance[18,19], and can deal with point mutations as well as structural variants. Networks have gained much attention in the past decade as an alternative method for studying prokaryotic evolution, including plasmids[3,8,18–20]. Plasmid gene-sharing networks have proven a useful means to track AMR and virulence dissemination, yielding deeper insights into HGT events[17,21,22]. However, the main drawback of previous work relying on plasmid sequence alignments is the exclusion of important non-coding elements, such as non-coding RNAs, promoter regions, CRISPRs (clustered regularly interspaced short palindromic repeats), stretches of homologous sequences, or putative, disrupted and currently unannotated genes. A more comprehensive approach could consider a plasmid network based on estimates of alignment-free sequence similarity[23]. Alignment-free genetic distance methods are becoming established tools for the analysis of large genomic datasets, and their usefulness has been validated in both prokaryotes and eukaryotes[19,23–26]. A recently published Plasmid ATLAS tool by Jesus et al.[27] provides an illustration of such an approach, with a network of plasmids constructed based on pairwise genetic distances estimated using alignment-free k-mer matching methods implemented in Mash[28].

In this work, we quantify the genetic similarity between more than 10,000 bacterial plasmids available on NCBI's RefSeq database and construct a network that reflects their relatedness based on shared k-mer content. Applying a community detection algorithm allows us to cluster plasmids with high genetic similarity into cliques (complete subgraphs) revealing a strong underlying population structure. We find cliques to be highly correlated with the gene content of the plasmid backbone, bacterial host and GC (guanine-cytosine) content, as well as replicon and MOB types. Uncovering the structure of the full plasmid population further allows the discovery of candidates for yet-undescribed replicon genes and provides insight into broad-scale plasmid dynamics. Taken together, our results illustrate the potential of network-based analyses of plasmid sequences and opens up the prospect of a natural, exhaustive classification framework for bacterial plasmids.

## Results

**A dataset of complete bacterial plasmids**. A dataset of complete bacterial plasmids was assembled comprising 10,696 sequences found in bacteria from 22 phyla and over 400 genera (Supplementary Data 1, Fig. 1a and Supplementary Fig. 1). The composition of plasmid hosts reflects current research interests, with the Proteobacteria and Firmicutes phyla together representing over 84% of plasmid sequences. The dataset includes plasmids from a diversity of bacterial hosts, with 66 plasmids from unknown bacterial families, 14 from uncultured bacteria and 37 samples from *candidatus* species (Supplementary Data 1). In total, 510,463 different coding sequences (CDSs) were identified in the plasmid dataset. In all, 66.01% of the CDSs were predicted to encode a hypothetical protein, 27.9% had a known product with Gene Ontology (GO) biological process annotation, with the remaining 6.09% encoding a known protein product with unknown biological function (Fig. 1b). There are 3,328,916 bacterial genes available in the RefSeq database (NCBI Gene Statistics accessed on 19 June 2019), meaning that roughly 1 in 20 of the currently known bacterial genes are plasmid borne. The GO biological processes associated with plasmid CDSs are diverse. After accounting for multiple occurrences of annotated CDSs in the dataset, the dominant associated GO terms relate to catabolic and biosynthetic processes (20.64% relative to total number of annotated CDSs), transposon mobility (17.09%) and positive and negative regulation of transcription (7.70%). Replicon-based typing classified 27.66% of the plasmids into 163 different replicon types (Fig. 1c and Supplementary Fig. 2). However, 31.67% of these classified plasmids were assigned to multiple replicon types. MOB typing was more comprehensive, successfully classifying 32.63% of the plasmids into six MOB types, of which 9.48% were assigned to multiple types (Fig. 1c). Unsurprisingly, classification by these two methods performed best for well-studied plasmids of the phyla Proteobacteria and Firmicutes.

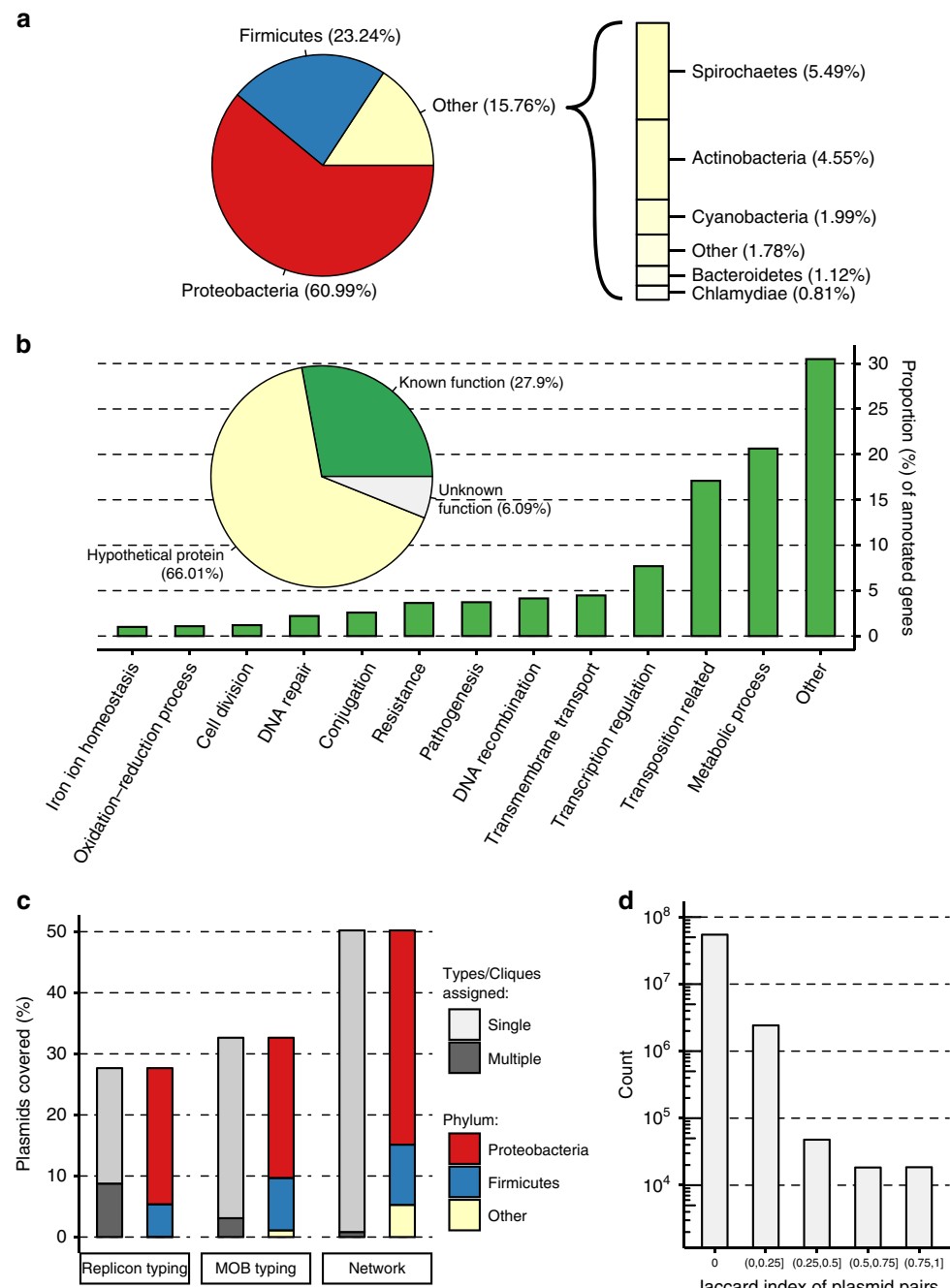

**Fig. 1 Summary of the dataset of complete bacterial plasmids. a** The distribution of host phylum represented in the plasmid dataset. **b** Functional annotation of plasmid-borne genes. The pie chart shows the proportion of unique CDSs with hypothetical function as predicted by Prokka[48], and CDSs (genes) with known/unknown biological function based on GO annotation. The bar chart provides the most common biological functions associated with plasmid-borne genes also considering the respective frequency of these genes on plasmid genomes. **c** The percentage of plasmids covered by the three classification methods: replicon and MOB-typing schemes, and clique assignment. **d** The distribution of pairwise plasmid similarities (Jaccard index).

**Uncovering the population structure of plasmids**. We constructed a network based on the plasmid pairwise sequence similarities. This represents a weighted, undirected network with plasmids (vertices) connected by edges indicating similarity (Supplementary Fig. 3). Similarity was scored using the exact Jaccard index (JI), defined as the size of the intersection divided by the size of the union of two sets of $k$-mers. Plasmid pairs that shared <100 $k$-mers were considered to have a JI equal to zero. This cut-off value was implemented since the majority of CDSs found on plasmids have lengths >100 bp, thus only a fraction of the functional genome is common between plasmids with low

shared $k$-mer count (Supplementary Figs. 4 and 5). The majority of plasmid pairs shared little to no similarity (Fig. 1d). In all, 6.14% (657) of the plasmids were singletons, while 3.31% (354) were connected to only one other plasmid, illustrating the high levels of diversity across bacterial plasmid genomes. It follows that plasmids with more $k$-mers in common are more likely to share the same functional genetic elements and hence participate in similar biological processes falling within the same host niche (Supplementary Fig. 5). Such plasmids are presumed to form cliques within the network with higher internal JI score. The objective is then to identify cliques that contain plasmids with

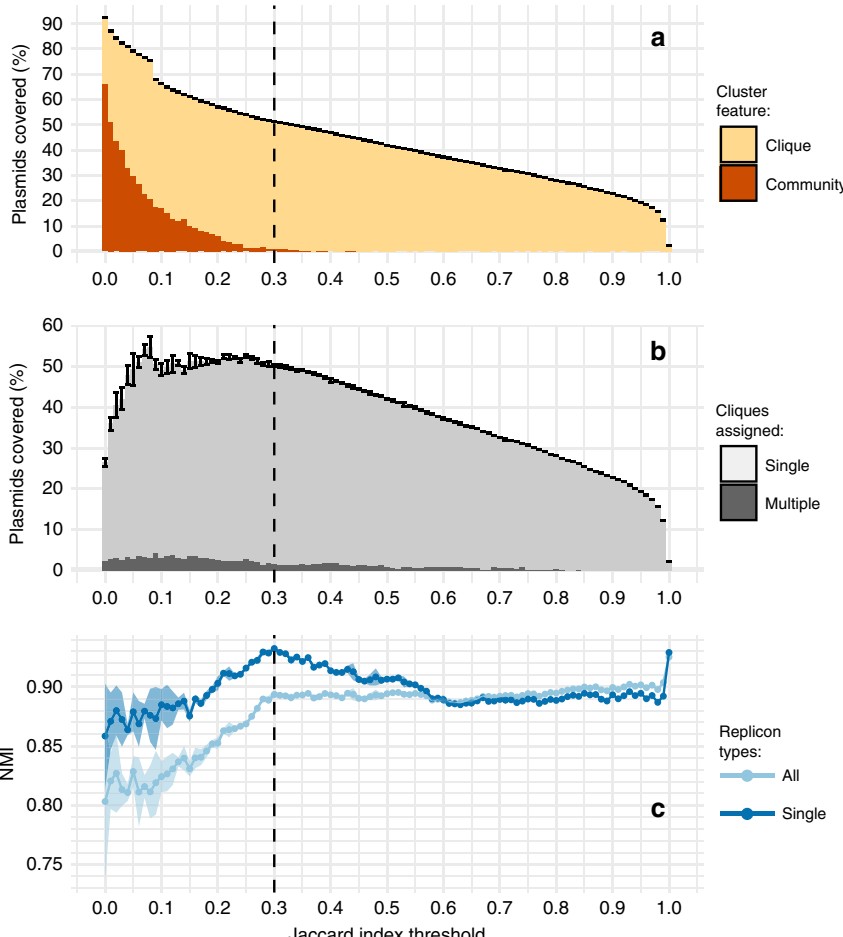

**Fig. 2 Optimization of OSLOM performance.** A range of Jaccard index (JI) thresholds were applied to the original plasmid network (Supplementary Fig. 3) with edges below a particular threshold being removed prior to OSLOM analysis. During the process, several criteria were considered: **a** clique to community ratio; **b** percentage of plasmids covered by the cliques; **c** the congruence with replicon typing measured by NMI score. NMI was calculated for all cliques containing plasmids assigned to a single or multiple replicon types (legend: All) and just to a single replicon type (legend: Single). Error bars (**a**, **b**) and light-coloured shading (**c**) provide ±2 standard deviations (SDs) of uncertainty. Standard deviation around every value on the y-axis across all JI thresholds assessed (points and bars) was calculated based on results of $n = 5$ iterations of the OSLOM software (see 'Methods'). The dashed vertical line indicates the selected optimal JI threshold of 0.3.

markedly higher similarity between themselves, relative to their immediate network neighbourhood.

Listing all cliques of our large plasmid network and assessing their internal similarity is computationally intractable with current tools[29]. A solution for a single clique can be quickly verified, but the time required to process all possible cliques scales rapidly as the size of the network increases. As an alternative solution, a stochastic community detection algorithm OSLOM (Ordered Statistics Local Optimization Method) was implemented[30]. OSLOM detects communities (i.e. densely interconnected subgraphs) with statistical significance, meaning that they have a low probability of being encountered by chance in a random network with similar features to the plasmid network. OSLOM is well suited for this task since it can be used to analyse undirected networks with overlapping communities. In addition, OSLOM shows similar performance to other widely used methods such as Infomap or Louvain[30,31], which, unlike OSLOM, were unable to analyse this dataset due to computational limitations. To validate the results from the stochastic clique assignment, all communities of size three or more detected by OSLOM were assessed for their completeness (i.e. whether they form cliques) against the original plasmid network (Supplementary Fig. 3).

Despite the notable dissimilarity among plasmids, the original network was too dense (network density = 0.0438) to yield a consistent performance for every OSLOM run (Fig. 2 and Supplementary Figs. 3 and 6). Furthermore, a large proportion of communities detected did not form cliques and would have to be disregarded (Fig. 2a). A JI threshold was introduced to increase the sparsity of the network and to upweight more similar plasmids, thus optimizing the performance of OSLOM. A range of thresholds were assessed based on the following criteria: (i) the clique to community ratio (Fig. 2a), (ii) the proportion of plasmids assigned to cliques (Fig. 2b), (iii) the congruence with replicon-based typing (Fig. 2c) and (iv) the consistency of OSLOM performance (Fig. 2 and Supplementary Fig. 6). The optimum threshold was consistently obtained at a JI of 0.3. The resulting sparse network is shown in Fig. 3 (network density = 0.00128).

The OSLOM-guided clique detection algorithm offers flexibility and identifies cliques of plasmids with a wide range of internal similarity scores (Supplementary Fig. 7). We assessed the importance of considering pairwise JI distances as a continuous variable by reanalysing the dataset with the Bron–Kerbosch Maxclique algorithm[32], implemented in the graph-tool Python library[33]. The Bron–Kerbosch algorithm is computationally

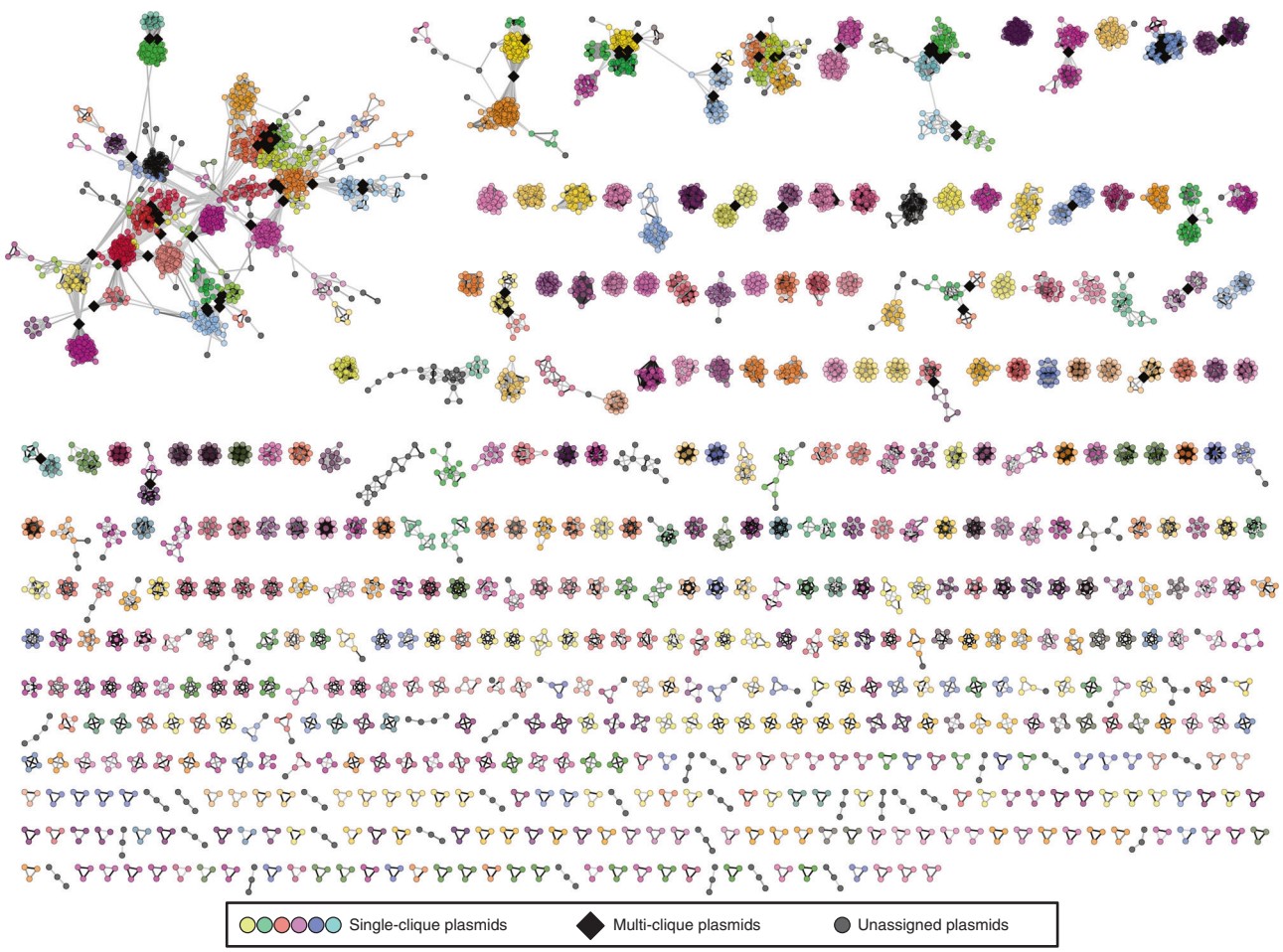

**Fig. 3 Sparse network of plasmids assigned to cliques by OSLOM algorithm (network density = 0.00128).** The network includes 5371 plasmids (nodes) assigned into 561 cliques (complete subgraphs). The completeness of identified cliques was evaluated based on the original network (Supplementary Fig. 3). 5008 unassigned plasmids, which formed disjoined singletons and pairs, were removed from the network. Coloured nodes indicate plasmids assigned to a single clique.

highly effective, but the pairwise distances between plasmids are treated as binary values defined by the given threshold. Applied across a range of JI thresholds, the Max-clique approach systematically identifies a very large number of cliques (Supplementary Fig. 8A), assigns a large proportion of plasmids to multiple cliques (Supplementary Fig. 8B) and leads to a low correlation between resulting cliques and plasmid replicon types (Supplementary Fig. 8C).

**Plasmid cliques agree with current typing schemes.** Analysis of the sparse network with OSLOM successfully assigned 50.21% (5371) of the plasmids into 561 cliques of size three or more (Figs. 1c and 3 and Supplementary Fig. 14). Only 1.64% (88) of these plasmids were assigned to multiple cliques, and these were found in the densest regions of the network and at the interfaces between cliques indicating the presence of 'chimeric plasmids' (i.e. hybrid plasmids generated through merging of two different plasmids), large-scale transposition or recombination events, or extensive repeated transposition/recombination (Figs. 1c and 3). In addition, this approach covered 564 plasmids from phyla other than the Proteobacteria and Firmicutes, namely from Spirochaetes, Chlamydiae, Actinobacteria, Tenericutes, Bacteroidetes, Cyanobacteria and Fusobacteria. Interestingly, after applying the 0.3 JI threshold, 38.01% (4066/10696) of plasmids that could not be assigned to cliques of size three or more were separated from

the network as singletons, while 10.10% (1080) shared an edge with a single plasmid. Therefore, only 1.67% (179) of plasmids were effectively left unassigned. Nonetheless, due to the apparent lack of shared genetic signal, plasmid singletons and pairs were not considered in subsequent analyses. To assess the extent to which 'mobile elements' shared between plasmids affect the classification into cliques, we repeated the clique assignment analyses after having removed all accessory CDSs (29,913) associated with transposition, pathogenesis, or resistance (Supplementary Fig. 9). Pruning these genes did not markedly affect the assignment of plasmids into cliques, which gives support to the genetic signal being driven by the genetic similarity of plasmid backbones rather than shared mobile genetic elements.

Clique purity and normalized mutual information (NMI) were used to assess the quality of clique-based classification (see 'Methods'). These metrics were calculated for cliques comprising plasmids with identified replicon type, plasmids carrying a single identified replicon type, or plasmids with assigned MOB type. Untyped plasmids were disregarded. The observed purity scores were high (>85%), indicating the homogeneity of cliques for a particular plasmid type (Supplementary Fig. 10). This was particularly the case for MOB types (purity = 0.9887) and plasmids assigned to a single replicon type (purity = 0.9522). NMI provides an entropy-based measure of the similarity between two classification systems where a score equal to one indicates identical partitioning into classes, while zero means

independent classification. NMI penalizes differences in the number of assignment classes, which justifies the low score observed when assessing clique-based versus MOB-based typing (NMI = 0.5223). Nevertheless, high NMI scores were obtained when considering a replicon-based classification scheme (NMI = 0.9044 all types, and NMI = 0.9336 for single replicon types). It follows that plasmids with the same replicon type often fall together within the same clique. This is also supported by the high correlation between the clique membership size and the number of plasmids assigned to the corresponding replicon class (Supplementary Fig. 11, $R^2 = 0.862$ for plasmids assigned to a single replicon types).

There are exceptions where plasmids from larger replicon classes are further resolved into a few smaller evolutionary-related cliques. One such example is provided by the 22 'broad-host-range' IncP plasmids, which have been split into three cliques (14, 118 and 332) (Supplementary Fig. 12, Supplementary Data 1). While plasmids within these cliques share notably high JI similarity, the similarities between cliques remain low. This is especially true for clique 332 and 14, for which between-clique similarity is zero. Interestingly, plasmids from clique 332 have been exclusively associated with Gammaproteobacteria, while the ones from cliques 118 and 14 are mostly found in hosts from the Betaproteobacteria class. This arrangement of IncP into multiple cliques with a more constrained host range is in line with previous findings of weaker incompatibilities in IncP[34] and the existence of multiple genetically distinct IncP sub-lineages whose backbone is coadapted to their host[35]. Another example of a genetically heterogeneous replicon type is provided by IncY and p0111 plasmids collected from *Escherichia coli* strains, which fall into three cliques (119, 230 and 372) (Supplementary Fig. 13). Clique 119 and 372 cluster IncY and p0111 plasmids, respectively, with a single, possibly misplaced IncFIB plasmid. Conversely, clique 230 comprises both IncY and p0111 plasmids, with a remarkably related genetic backbone. The latter result raises questions on the distinctiveness of IncY and p0111 plasmid types.

**Candidate replicon genes recovered from untyped plasmids**. The majority of plasmids with unknown replicon types formed small cliques (Supplementary Fig. 14). In fact, 81.02% of the smallest cliques (carrying three to five plasmids) contain exclusively untyped plasmids. Together with the aforementioned singletons and lone plasmid pairs, this trend highlights the many understudied and underrepresented plasmids in sequence databases. Accordingly, the next objective was to investigate the genetic content of untyped cliques to determine candidate replicon genes and further traits of biological relevance.

In total, there are 388 cliques with no assigned replicon types. As the cliques tend to be homogeneous for a replicon type, only the core genes (i.e. genes occurring on all plasmids of a particular clique) found on untyped cliques were considered. Core genes were translated into protein sequences and screened against the translated PlasmidFinder database using TBLASTN[36]. A range of *e* values were assessed to determine the threshold maximizing the discovery of replicon candidates while minimizing false positives (Supplementary Fig. 15). The majority of plasmids were assigned to one replicon type with some plasmids having hits to a maximum of three to four different types. Accordingly, the optimal *e* value threshold was selected when the total number of core gene hits started to diverge from the number of untyped cliques covered. A conservative *e* value threshold of 0.001 was chosen, which resulted in the identification of 105 candidate genes from 106 plasmid cliques. The accession numbers and positions of candidate genes are listed in the Supplementary Data 1 (*Candidate_replicon_gene* column) for all carrier plasmids.

To verify the plausibility of the identified gene candidates, HMMER (version 3.2.1) was used to scan amino acid sequences for known protein domain families found in the Pfam database (version 32.0)[37]. One hundred and sixty-six families, with *e* values lower than 0.001, were identified on 97 protein sequences and were most commonly associated with replication initiation (Supplementary Fig. 16). Moreover, the majority of functions associated with the discovered protein families relate to plasmid replicon proteins. For example, domains with helix–turn–helix motifs are important for DNA binding of replicon proteins and allow some proteins to regulate their own transcription[38]. Other examples of transcriptional regulators also exist in plasmid replicon regions, while DNA primase activity has been found on the RepB replicon protein[38]. Interestingly, replicon proteins involved in rolling-circle replication (a mechanism of plasmid replication) share some of their motifs with proteins involved in plasmid transfer and mobilization[38]. This could explain why some of the discovered domain families are linked to plasmid mobilization. On the whole, the candidate replicon genes are highly specific to a particular clique of plasmids and should assist description of new incompatibility types.

**Cliques exhibit common GC content and bacterial hosts**. The unprocessed plasmid network exhibited a pronounced structure in terms of the plasmid nucleotide composition, measured by GC content (Supplementary Fig. 3). This trend was also reflected in the clique composition (Supplementary Fig. 17A). Within a clique, the standard deviation of GC content rarely exceeds 0.02 and is weakly correlated with the clique size ($R^2 = 0.0155$) (Supplementary Fig. 17B). Moreover, a significant difference in GC content is often found between cliques. Analysis of variance, followed by a Tukey's test, found that 85.3% of the time the GC content between two cliques differs significantly (adjusted *p* value < 0.001). In contrast, the sequence lengths of plasmids within a clique are more variable, but are also not strongly correlated with clique size ($R^2 = 0.029$) (Supplementary Fig. 17C, D). Similarly, a Tukey's test showed that a significant difference in plasmid length between cliques is observed <34% of the time (adjusted *p* value < 0.001).

Plasmid GC content has been shown to be strongly correlated to the base composition of the bacterial host's chromosome[39]. Indeed, the cliques showed a very high homogeneity (purity) relative to their hosts (Supplementary Fig. 18), a trend that has been identified in other plasmid network reconstruction efforts[21]. At higher taxonomic levels, cliques have near-perfect purity scores (>0.99). The purity score slightly decreases at the level of the plasmid host family, reaching a value of 0.807 at the species level. Therefore, plasmids with high genetic similarity rarely transcend the level of the bacterial genus, which suggests a limited host range for the vast majority of plasmids. However, these results need to be carefully considered due to inherent biases in the dataset, especially in terms of the predominance of well-studied taxa. Overall, the plasmid cliques show a strong intrinsic propensity towards confined GC content and are found in a limited range of bacterial hosts.

**Plasmids within cliques have uniform gene content**. The gene content of cliques was assessed for all genes occurring five or more times in the dataset. This threshold was chosen to facilitate computation, and to adequately characterize more prevalent genes. In total, 15,851 out of 35,883 (44.17%) of the assessed genes were 'core' genes, meaning they had a within-clique frequency equal to one, suggesting an overall uniformity of gene content in cliques (Supplementary Fig. 19). Furthermore, 6577 (18.33%) of the genes were 'private'. Private genes are those found

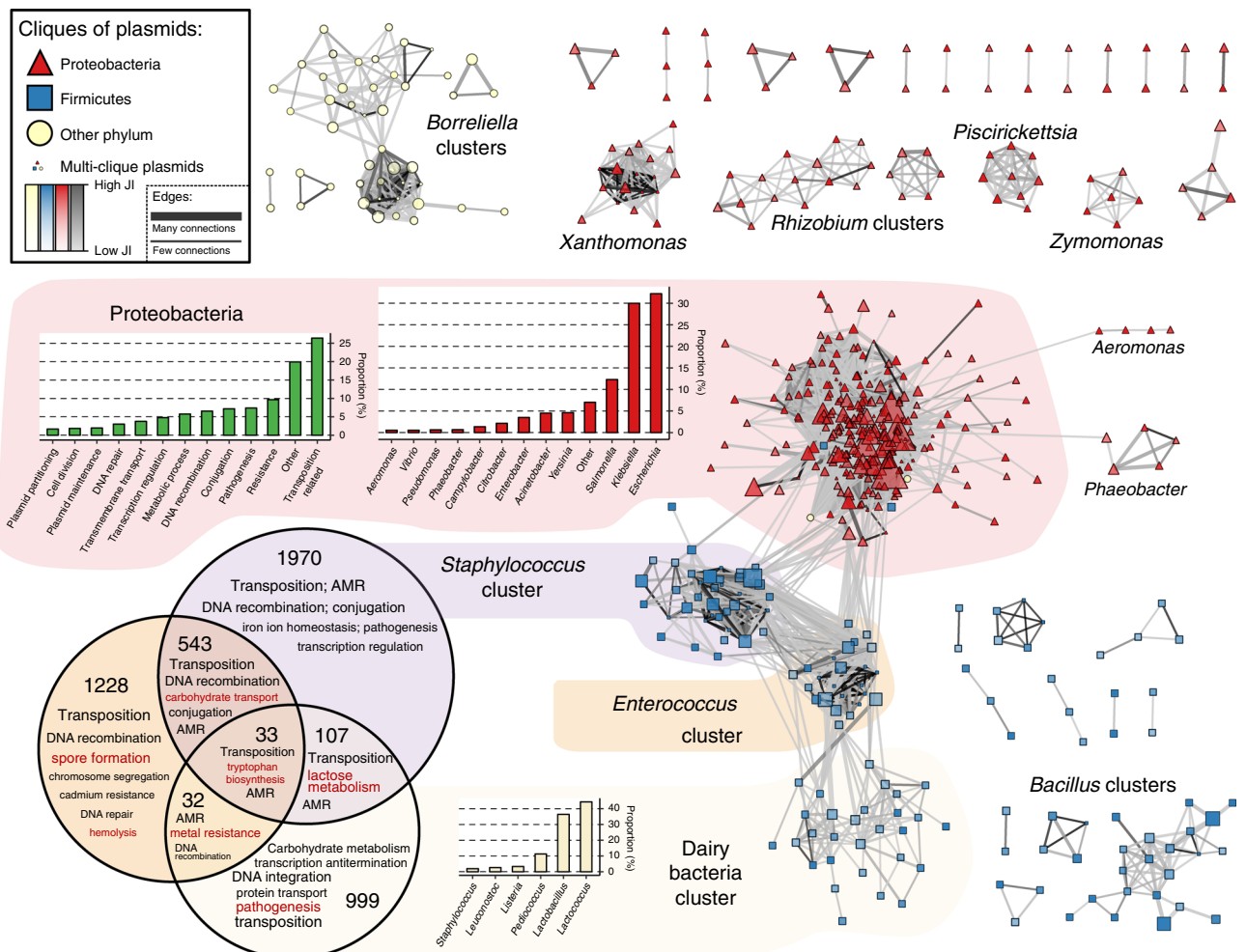

**Fig. 4 The network of cliques.** Cliques, represented as vertices, are connected with an edge if the average Jaccard index (JI) between plasmids of two cliques is >0.01. The colour of the edges indicates the average JI, while the width is proportional to the number of connections between a pair of cliques. The shape and colour of the cliques indicates the phylum of the predominant bacterial host. The size and the transparency are proportional to the clique size and the internal JI, respectively. The cliques form multiple clusters, which have been named based on the genus of the bacterial host characteristic for a particular cluster. There are two exceptions—the Proteobacteria and the Dairy (Lactic) cluster whose respective genera distributions have been provided. The most common GO biological functions of the genes found on plasmids of Proteobacteria, *Staphylococcus*, *Enterococcus* and Dairy clusters were further assessed. During the assessment, the respective frequencies of the genes were considered. In case of Proteobacteria, the bar chart distribution of the biological functions is provided. The shared and core gene content of *Staphylococcus*, *Enterococcus* and Dairy clusters is presented in the Venn diagram with the numbers in the diagram indicating the number of core and shared genes.

in only one clique, with a frequency of one, and their relatively high abundance in the dataset suggests the uniqueness of some cliques with respect to their gene content. However, there is an inherent bias. Plasmids within larger cliques tend to be more dissimilar and share proportionally fewer genes (Supplementary Fig. 20). This pattern can, in part, be explained by the broader gene content of large cliques and the high sequence similarity required for same-gene clustering (95%) within the default implementation of the Prokka–Roary annotation pipeline. In all, 31.94% of cliques containing five or more plasmids were found to have 1 to 10 core genes. However, cliques exhibited a wide range in the number of core genes with 7.74% of cliques carrying over 100 shared genes. Interestingly, 13.55% (42) of cliques had no core genes that could also be an artefact of the gene annotation pipeline sensitivity or poor-quality assemblies. For instance, plasmids from 19 cliques carried no recognized genes from the pool of 35,883 assessed genes. Functionally, core genes were found to be more often associated with various metabolic processes, transcription regulation and transmembrane transport

(Supplementary Fig. 21) when compared to the overall distribution of GO terms, shown in Fig. 1b. Similarly, fewer core genes were involved in transposon movement, pathogenesis and resistance.

**Inferring HGT through clique interactions.** Gene content was also considered in the context of clique structure and interconnectedness. To do so, the original network of plasmids (Supplementary Fig. 3) was rearranged such that: (i) plasmids assigned to the same clique were clustered under a single vertex; (ii) plasmids assigned to multiple cliques were left as solitary vertices anchoring the cliques; (iii) unassigned plasmids were removed. The resulting network is shown in Fig. 4. As highlighted earlier, large cliques generally show lower internal similarity compared to the smaller ones. It is important to note that an arbitrary JI threshold of 0.01 was introduced in Fig. 4 to assist visual interpretation, but the unfiltered version of the network is provided in Supplementary Fig. 22.

The clustering of cliques in Fig. 4 shows high concordance with the phylogenetic hierarchy of the bacterial hosts. On a global scale, there are four large interconnected clusters (three corresponding to cliques from the phylum Firmicutes and one from the Proteobacteria), eight disjointed clusters and a dozen singled-out triplets and pairs. The clique clusters mostly contain plasmids from a specific genus with some minor deviations—hence the cluster naming. The only two exceptions are the large and diverse Proteobacteria cluster, which harbours plasmids mainly from the genera *Escherichia*, *Klebsiella* and *Salmonella* and the Dairy bacteria. The majority of genes identified in these four large clusters were those functionally involved in transposition. Specifically, 26.4% of the genes in the Proteobacteria cluster were transposition related. In addition, 9.66% of the genes in the Proteobacteria were involved in some form of AMR or metal resistance, and 7.38% in pathogenesis, which may reflect the high number of pathogens found in this phylum[40].

The core and shared gene content of the three Firmicutes clusters (*Staphylococcus*, *Enterococcus* and Dairy) was also assessed (Fig. 4, Venn diagram). Gene sharing was most common between the clusters associated with *Staphylococcus* and *Enterococcus* potentially indicating a high frequency of HGT between them, and the least between the *Staphylococcus* and Dairy bacteria cluster. Analysing the content of these shared genes provides insight into both plasmid function and dynamics, such as the identification of HGT events. For example, the same lactose metabolism genes were found in both *Staphylococcus* and Dairy bacteria plasmids. Also, the *trpF* gene, involved in tryptophan biosynthesis and previously associated with the Tn*3000* and Tn*125* transposable elements[41,42], was found on plasmids in all three clusters. In contrast to these, the more disjoint clusters of plasmid cliques observed for other genera may be driven by the species' ecology and life history, which may lead to limited opportunities for contact between lineages. Such an explanation seems plausible for strict pathogens with restricted host range, such as *Xanthomonas* or *Borrellelia*. Conversely, for lineages with a wider environmental niche like *Bacillus*, the lower connectivity between cliques may be due to intrinsic genetic factors leading to lower between-plasmid recombination and/or transposition rates.

## Discussion

Using alignment-free sequence similarity comparison and subsequent network analysis, we uncovered strong population structure in bacterial plasmids. This approach, applied to a comprehensive set of complete bacterial plasmids, yielded a network in which over half of the plasmids were classified into cliques. There is a significant improvement in coverage over existing plasmid typing methods. Additionally, the cliques capture biologically meaningful information. For example, plasmids assigned to the same clique show good accord with replicon and MOB-typing schemes, high homogeneity in terms of their respective bacterial hosts and similar GC and gene content.

A network-based representation of plasmid sequence similarities condenses both vertical and horizontal evolutionary histories in a similar fashion to gene-sharing networks[17,21,22], making it ideally suited for the identification of mobile genetic elements. The cliques we recovered delineate clusters of plasmids with shared evolutionary history. This in turn allows for inference on the nature of HGT events and plasmid function. Moreover, the approach facilitates identification of new replicon gene candidates, as well as detailed investigation of the distribution of plasmid-borne genetic determinants of incompatibility, MOB, AMR, virulence and transposon carriage. Such meta-information

could be incorporated within the network framework owing to a plethora of well-maintained bioinformatics tools, ever-growing genetic databases and GO efforts to systematize gene annotation.

The strong, host-constrained population structure we document for the majority of bacterial plasmids points to transposable genetic elements as the main drivers of HGT in bacteria. In such a case, plasmids primarily act as vehicles for the transfer of genetic material between different bacterial taxa and are eventually lost, while transposons are more successful at maintaining themselves by relocating onto a host-compatible plasmid or the chromosome. Such dynamics could explain the relative uniformity of plasmid cliques in their host range, gene and GC% content, as well as the excess of transposable elements shared between cliques of different taxa. This likely extends to the so-called 'broad-host-range' plasmids such as IncP, whose representatives in our dataset fell into three genetically distinct cliques associated to different host species.

JI (i.e. the fraction of shared *k*-mers) was chosen as a measure of sequence similarity between pairs of plasmids due to it being a straightforward metric, which considers genome sequences as a whole, embodying both point mutations and large-scale genome rearrangements. As a result, it is not biased by the ability to annotate genes, open reading frames, or other genetic elements. In addition, it is not prone to errors and biases intrinsically associated with alignment-based methods, such as: a priori assumptions about the sequence evolution, higher inaccuracy when comparing more dissimilar sequences, or suboptimal alignments[23]. JI can in principle provide fine-scale resolution when comparing small genomes, a characteristic common to the majority of plasmids. Conversely, JI is sensitive to varying genome sizes[28] and plasmids are known to differ more than 1000-fold in sequence length[7,43]. While differences in plasmid genome size can lead to a drop in JI score even when high proportions of *k*-mers are shared, sequence length variation did not seem to impact our structuring into cliques, which comprise plasmids of different lengths (Supplementary Fig. 17C, D).

Assessing the statistical significance of all cliques is computationally intractable given the size of the network. Hence, OSLOM community detection algorithm was employed to uncover cliques of plasmids with high genetic similarity. In an effort to optimize the performance of the OSLOM algorithm and maximize the number of biologically meaningful cliques, all edges with a JI value <0.3 were removed from the network prior to the analysis. This threshold was chosen to maximize compliance with replicon-based typing as well as several other criteria. While our classification of plasmids into cliques is fairly robust to this exact JI threshold, we appreciate that a 0.3 JI threshold remains somewhat arbitrary. This being said, any taxonomy based on sequence similarity will be partly subjective. As such, our 0.3 JI threshold is comparable in its subjectivity to the 95% average nucleotide identity, which was set over a decade ago and is routinely used to define species boundaries in prokaryotes[44]. However, depending on the question pursued, enforcing a strict JI threshold may not be necessary, and it could be left to plasmid sequences in the network to solely inform the cut-offs. Some boundaries are likely to be blurrier than others, largely reflecting the extensive variation of genetic inheritance in different bacterial hosts.

Our results suggest it should be possible to devise a 'natural', global sequence-based classification scheme for bacterial plasmids. This being said, our findings do not diminish the relevance of replicon and MOB-typing schemes, rather they build upon these prior classification schemes and may extend them to plasmids from understudied and uncultured bacteria. Beyond just plasmid classification, our network-based approach also has the potential to infer key features of plasmid groupings. Indeed, plasmid clique assignment can be completely automated and

inspection of any particular area of the network facilitates biological inference about plasmid dynamics and their biological features within various groups of bacterial hosts.

## Methods

**Assembling a dataset of complete bacterial plasmids.** A dataset of complete plasmids was downloaded from NCBI's RefSeq release repository[45] on 26 September 2018. The metadata accompanying each plasmid sequence was parsed from the associated GenBank files. The resulting dataset was then systematically curated to include only those plasmids sequenced from a bacterial host and with a sequence description, which implies a complete plasmid sequence (regular expression term used: 'plasmid.*complete sequence'). This is a simpler, but similar approach to a previously reported curation effort by Orlek et al.[9]. Nevertheless, a large portion of unsuitable entries, such as gene sequences, partial plasmid genomes, whole genomes, non-bacterial sequences and other poorly annotated sequences, were removed. The final dataset included 10,696 complete bacterial plasmids as listed with full metadata in Supplementary Data 1.

Information about the taxonomic hierarchy of plasmid bacterial hosts was obtained with the *ncbi_taxonomy* module from the ETE 3 Python toolkit[46]. To determine the replicon and MOB types of plasmids included in the dataset, we used the PlasmidFinder replicon database (version: 2018-09-04)[12] and the MOBtyping software[47]. The PlasmidFinder database was screened using nucleotide BLAST[36] with a minimum coverage and percentage identity of 95%. In cases where two or more replicon hits were found at overlapping positions on a plasmid, the one with the higher percentage identity was retained. For determining the plasmid MOB type, MOBtyping software was used with the recommended settings of 14 PSI-BLAST iterations.

Plasmid CDSs were annotated using Prokka[48] (version 1.13.3) and Roary[49] (version 3.12.0) pipelines run with default parameters. The identified CDSs were further associated with GO terms[50,51] to facilitate downstream gene content analysis. Since Prokka uses a variety of databases to annotate identified CDSs, different resources have been used to append the corresponding GO terms. For example, GO terms for CDSs with a known protein product have been obtained using Uniprot's 'Retrieve/ID Mapping' tool[52], while the GO terms for CDSs with just the HAMAP family were obtained with the hamap2go mapping table[53] (version date: 2019/05/04). CDSs annotated with the ISfinder database were given GO terms GO:0070893 and GO:0004803 in order to associate them with transposition. Similarly, CDS annotated with Aragorn, MinCED and BARRGD were given GO:0006412, GO:0099048 and GO:0046677 terms, respectively.

**Assessing similarity between pairs of plasmids.** The exact JI was used as a measure of similarity between all possible plasmid pairs. Each plasmid sequence was converted to a set of 21 bp *k*-mers. The JI was then calculated as the fraction of shared *k*-mers between two sets. JI thus takes a value between 0 and 1, where 1 indicates 100% *k*-mer similarity and 0 indicates no *k*-mers shared. This allows balanced comparison of diverse plasmid genomes and universality. Also, JI does not weight *k*-mers based on their abundance, like the popular $D_2^*$ and $D_2^S$ statistics[54], which would exacerbate the inherent sampling biases towards well-studied species to the dataset. We applied Bindash[55] to calculate the exact JI, which resulted in the creation of a plasmid adjacency matrix, which was used to build the network. All networks presented here have been explored and visualized using the Cytoscape software[56].

**Implementing OSLOM community detection algorithm.** OSLOM (version 2.5) was applied to identify cliques (complete subgraphs) with high internal JI similarity in the plasmid network[30]. OSLOM aims to identify highly cohesive clusters of vertices (communities) that may or may not be cliques (complete subgraphs). The statistical significance of a cluster is measured as the probability of finding the cluster in a configuration model, which is designed to build random networks while preserving the degrees (number of neighbours) of each vertex. The method locally optimizes the statistical significance with respect to vertices directly neighbouring a particular cluster. In brief, OSLOM starts by randomly choosing vertices from a network that is regarded as clusters of size one. These small clusters alongside their neighbouring vertices are assessed. Vertices are scored based on their connection strength with a particular cluster and are either added or removed from the cluster. The process continues until the entire network is covered. Due to the stochastic nature of the algorithm, this network assessment goes through many iterations, after which the frequently emerging significant clusters (i.e. communities) are kept. The algorithm then proceeds to assess the clusters of the next hierarchical level; vertices belonging to the significant clusters are condensed into super-vertices with weighted edges connecting them. The process of cluster assessment is repeated at higher hierarchical levels until no more significant clusters are recovered.

OSLOM was executed for an undirected and weighted network with the following parameters:
```
oslom_undir -w -t 0.05 -r 50 -cp 0 -singlet -hr 0 -seed 1.
```
Clusters were considered significant if their *p* value was lower than 0.05 (`-t 0.05`). The number of iterations required before the recovery of significant clusters was set to 50 during the search for the optimally sparse network (`-r 50`), and 250 for the final network analysis after the introduction of the 0.3 JI threshold

(`-r 250`). After the iteration process, OSLOM considers merging similar significant clusters if the significance of their union is high enough. This feature can potentially yield less cliques and was suppressed with the coverage parameter set to zero (`-cp 0`), thus forcing OSLOM to opt for the biggest and most significant cluster from a set of similar clusters. In addition, OSLOM tries to place all vertices of a network in a cluster, which is also unfavourable for clique recovery and was suppressed with option `-singlet`. Lastly, cliques can only be recovered at the first hierarchical level. Therefore, the OSLOM analysis of the higher hierarchical levels was disregarded (`-hr 0`).

As mentioned earlier, OSLOM is a non-deterministic algorithm and the initial single-vertex clusters are chosen at random. While looking for the optimally sparse network, five OSLOM runs were executed to assess every JI threshold and were given seeds for a random number generator (`-seed`) of 1, 5, 42, 93 and 212. The final network analysis was performed with a seed equal to 42, after which only cliques were considered, with non-complete communities disregarded.

**Scoring NMI and purity.** The compliance of cliques with replicon and MOB-typing schemes was assessed by measuring the NMI and purity between them. NMI is a commonly used method to assess the performance of clustering algorithms[57]. For the two clustering/classification schemes ($C_1$ and $C_2$) NMI is defined as[58]:

$$\mathrm{NMI}(C_1, C_2) = \frac{I(C_1, C_2)}{\frac{[H(C_1) + H(C_2)]}{2}}. \tag{1}$$

In Eq. (1), the mutual information, also known as the information gain and denoted as $I(C_1,C_2)$, is an information theory concept that measures the reduction of uncertainty around $C_1$ given the knowledge about $C_2$, and vice versa. It is normalized by the averaged Shannon entropy ($H$) between $C_1$ and $C_2$. Shannon entropy tends to be larger as the number of classes in $C_1$ or $C_2$ approach the size of the dataset in question. Consequently, the NMI is sensitive to differences in the number of classes between $C_1$ and $C_2$, and to extensively fragmented classifications. The NMI equals one if the two classifications yield identical partitioning of the dataset, whereas a value of zero indicates complete incoherence. The NMI was measured using the *R* package *NMI* (version 2.0; https://CRAN.R-project.org/package=NMI). During the assessment, plasmids that were not classified by replication or MOB-typing schemes were disregarded.

Purity was used to estimate the homogeneity of cliques for replicon or MOB types, and plasmid host taxa. For a set of cliques C, and a plasmid typing scheme T, purity is defined as:

$$\mathrm{purity}(C, T) = \frac{1}{N} \sum_{c_i \in C} \max_{t_j \in T} \left| c_i \cap t_j \right|, \tag{2}$$

where N is the total number of plasmids covered by a set of cliques, C = { $c_1$, $c_2$, …, $c_i$ } is a set of cliques in which plasmids were placed, and T = { $t_1$, $t_2$, …, $t_j$ } are the types associated with plasmids. Similar to NMI, the purity scores a value between 0 and 1, with high purity indicating high homogeneity of classes in the dataset for a given set of plasmid types. The purity was only assessed for cliques that contain at least one typed plasmid. Untyped plasmids found within the assessed cliques were disregarded.

The full MOB, incompatibility types and clique assignments for each plasmid are provided in Supplementary Data 1.

**Reporting summary.** Further information on research design is available in the Nature Research Reporting Summary linked to this article.

## Data availability
The sequences that make up the dataset analysed during the current study are available in the NCBI's RefSeq repository (ftp://ftp.ncbi.nlm.nih.gov/refseq/release/plasmid/). The accession numbers of the sequences are listed in Supplementary Data 1. The source data underlying the figures are provided as a Source Data file.

## Code availability
All code used in this research is available at github (https://github.com/macman123/plasmid_network_analysis).

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

## Acknowledgements

M.A. was supported by a Ph.D. scholarship from University College London. L.v.d. and F.B. acknowledge financial support from the Newton Fund UK-China NSFC initiative (MRC Grant MR/P007597/1) and a Wellcome Institutional Strategic Support Fund (ISSF3) – AI in Healthcare (19RX03). F.B. additionally acknowledges support from the BBSRC GCRF scheme and the National Institute for Health Research University College London Hospitals Biomedical Research Centre. All authors acknowledge UCL Biosciences Big Data equipment grant from BBSRC (BB/R01356X/1). The funders had no role in study design, data collection, interpretation of results, or the decision to submit the work for publication. Lastly, we would like to thank Prof. Chris Barnes for providing advice on the network analyses.

## Author contributions

M.A. and F.B. conceived the project and designed the experiments. M.A. performed all the analyses under the guidance of L.v.D and F.B. J.M.S advised on plasmid biology. M.A., L.v.D. and F.B. take responsibility for the accuracy and availability of the results. L.v.D. provided moral support to M.A. M.A. wrote the paper with contributions from L.v.D. and F.B. All authors read and commented on successive drafts and all approved the content of the final version.

## Competing interests

The authors declare no competing interests.
