## [Peer Review File · Nature Communications]

Reviewers' comments:

Reviewer #1 (Remarks to the Author):

In this manuscript, Acman et al. systematically analysed the phylogenetic relatedness among 10696 bacterial plasmid sequences using a network approach based on k-mers, without using multiple sequence alignment. Similar approaches have been adopted to infer phylogenetic relatedness among thousands of complete prokaryote genomes as networks (<https://doi.org/10.1128/mSystems.00257-18>), and the biological relevance of k-mers in phylogenomic networks (e.g. <http://dx.doi.org/10.12688/f1000research.10225.2>) and in inferring phylogenetic relationships (e.g. Zielezinski et al. 2019 Genome Biology) are well-documented in the literature. Therefore, the use of k-mers in assessing plasmid evolution is appropriate. I am really glad to see k-mers are gaining momentum in phylogenetic research!

The novelty of this work lies in the use of k-mers in an innovative network approach for analysing plasmid genomes. The results from this work reveal correlation of k-mer similarity with gene content and GC content (as one would expect), and more interestingly, a strong correlation of k-mer similarity to bacterial host range and to the distinct type classifications.

The manuscript is very well-written and well-polished. The adopted methodology is refreshing, and appears appropriate to address the specific research questions. The use of OSLOM to identify statistically significant cliques is interesting and established. The calculations of the NMI and purity scores are scientifically sound, and the use of these statistics are appropriate. My assessment here largely focuses on the bioinformatic and network analyses of this work. I only have few minor comments, which I note below.

1. Lines 62-66: the application of k-mer approaches in phylogenetics extends beyond plasmid genomes. To make this manuscript stronger, and more convincing (particularly, to the skeptics of alignment-free methods among the research community), I recommend the authors to provide a little more background/context in this regard. Additional citations to review articles (e.g. <https://doi.org/10.1093/bib/bbx067> and <https://doi.org/10.1146/annurev-biodatasci-080917-013431>), or some of the references I highlighted in the first paragraph above, would be very helpful. This is really to emphasise that these methods are proven and tested in various data types, and not just limited in their applications in plasmid genomes.

2. L87 onward: to which is the 20.64% relative? Total proteins, or total annotated GO BP terms? This should be clarified.

3. L118-130: This section justifying the JI threshold of 0.3 is important. I think Figure 2 may be too technical as a main figure; the authors may consider presenting this figure as a supplementary figure instead. Also, the current Supplementary Figure S3 is quite pixelated – a larger, higher-quality figure would be very helpful.

4. L208: "The gene content of cliques was assessed for all genes occurring five or more times in the dataset." A brief justification of this approach (i.e. why five?) would be great. If this follows an earlier study, it should be clear. Also, how does this approach cope with gene duplication?

5. L323-327: The use of Jaccard index to assess the similarity of pairwise k-mer profiles is interesting and straightforward. It is simpler than the other methods, e.g. variants of D2 statistics (<https://doi.org/10.1089/cmb.2009.0198> and <https://doi.org/10.1089/cmb.2010.0056>) that incorporate normalization measures e.g. by the probability of specific k-mers found in the dataset. I think the choice of Jaccard index over other k-mer methods should be explained and justified in the text.

6. L352: the parentheses around "-singlet" are not necessary.

7. L358-359: "after which only cliques were considered with non-complete communities disregarded." I believe a comma between "considered" and "with" here would make the expression clearer?

Thanks for the opportunity to review this paper,
Cheong Xin Chan

Reviewer #2 (Remarks to the Author):

Comments to "Large-scale network analysis captures biological features of bacterial plasmids" by Acman et al.

1. Acman et al. performed network analysis of bacterial plasmids in database based on their shared k-mer content. They successfully classified almost half of the plasmids in database into cliques. They argued that their analyses could provide biological features of plasmids. While the present study is of interest in the research field of plasmid biology, however, this study mainly focused on the methodology of classification. The new findings in the present study are not necessarily clearly shown. If their analyses could classify more plasmids than before, it might be better to show the example(s) of newly classified plasmid(s) more in detail.
2. Many plasmids carried accessory genes with transposons. This reviewer could not understand the effects of them on the classification when the same accessory genes or transposons were carried by different plasmids.
3. They did some comparisons with the classifications based on Rep and MOB typing. Some plasmids even in the same Inc group were not necessarily classified into one clique (e.g. IncP1=IncP and IncP6=IncU), while other plasmids in the same Inc group were into a single clique (e.g. IncL/M., IncQ1=IncP-4). It might be helpful if they explain the reason(s). Related to this, there should be more information for Rep typing, such as IncP-7, IncP-9, and PromA.
4. They did not use plasmid sequences of uncultured (or unidentified) hosts.
5. Lines 171-173 and Supplementary Table 1. This reviewer could not understand the candidate genes in the table. Were they listed in 'Replicon_candidate'?
6. Lines 188-189 and Supplementary Figure 3. This reviewer could not understand this part probably because the colour gradient of the nodes in the figure was not clearly shown.

Reviewer #3 (Remarks to the Author):

This manuscript describes a method of identifying patterns in the similarity network of plasmids via the identification of cliques.

With respect to the network analysis aspect of the work (ignoring the biological relevance of the results), some problems can be identified:

1. The authors propose the use of the OLSOM community detection algorithm to identify cliques, arguing that direct maximal clique identification (max-clique) would be prohibitive computationally, given the NP-complete nature of the problem. However, this is incorrect. Although it's true that max-clique is NP-complete in the worst case, it is fact polynomial in many instances of the

problem. In particular, for sparse networks the problem tends to be tractable with the Bron-Kerbosh algorithm, which often runs close to linearly on the number of nodes, and thus usable for networks with up to millions of nodes. For the plasmid network with around 5K nodes, the full enumeration of maximal cliques should take a matter of seconds on a typical laptop. Free implementations of the max-clique algorithm are available in many packages (e.g. in the graph-tool Python library: https://graph-tool.skewed.de/static/doc/topology.html#graph_tool.topology.max_cliques)

It is also important to note that the community detection problem (e.g. via maximization of modularity, or the inference of the stochastic block model) are also NP-Hard in general (but also feasible in many instances), which renders the argument in the manuscript moot. (However, regardless of its feasibility, community detection is not the appropriate ansatz if the objective is to find cliques.)

2. The similarity network is converted into a binary network via thresholding. Although the authors discuss briefly the chosen threshold value, its choice is likely to have a very strong impact on the existence of cliques in the resulting network. Therefore, it is absolutely necessary to evaluate the robustness of the results on the choice of the threshold value.

The above points are serious problems with the current analysis, and it is imperative they are addressed before the the manuscript is further considered for publication.

Response to Reviewers

We would like to thank the Editor for the chance to review the manuscript and are grateful to the reviewers for their thorough assessment and useful comments.

A point-by-point response to all comments is included below. Our responses can be found immediately under each of the reviewer's comments which are shown in blue. Changes we have made in the main text and supplementary materials are shown in bold.

We hope that with these changes the manuscript is now suitable for publication. Thank you for your time and consideration.

Yours sincerely,

Mislav Acman and Francois Balloux, on behalf of all the co-authors

Reviewer #1 (Remarks to the Author):

In this manuscript, Acman et al. systematically analysed the phylogenetic relatedness among 10696 bacterial plasmid sequences using a network approach based on k-mers, without using multiple sequence alignment. Similar approaches have been adopted to infer phylogenetic relatedness among thousands of complete prokaryote genomes as networks (<https://doi.org/10.1128/mSystems.00257-18>), and the biological relevance of k-mers in phylogenomic networks (e.g. <http://dx.doi.org/10.12688/f1000research.10225.2>) and in inferring phylogenetic relationships (e.g. Zieleszinski et al. 2019 Genome Biology) are well-documented in the literature. Therefore, the use of k-mers in assessing plasmid evolution is appropriate. I am really glad to see k-mers are gaining momentum in phylogenetic research!

The novelty of this work lies in the use of k-mers in an innovative network approach for analysing plasmid genomes. The results from this work reveal correlation of k-mer similarity with gene content and GC content (as one would expect), and more interestingly, a strong correlation of k-mer similarity to bacterial host range and to the distinct type classifications.

The manuscript is very well-written and well-polished. The adopted methodology is refreshing, and appears appropriate to address the specific research questions. The use of OSLOM to identify statistically significant cliques is interesting and established. The calculations of the NMI and purity scores are scientifically sound, and the use of these statistics are appropriate. My assessment here largely focuses on the bioinformatic and network analyses of this work. I only have few minor comments, which I note below.

We are grateful to the reviewer for their positive assessment of the manuscript and recognizing the potential of our work.

1. Lines 62-66: the application of k-mer approaches in phylogenetics extends beyond plasmid genomes. To make this manuscript stronger, and more convincing (particularly, to the skeptics of alignment-free methods among the research community), I recommend the authors to provide a little more background/context in this regard. Additional citations to review articles (e.g. <https://doi.org/10.1093/bib/bbx067> and <https://doi.org/10.1146/annurev-biodatasci-080917-013431>), or some of the references I highlighted in the first paragraph above, would be very helpful. This is really to emphasise that these methods are proven and tested in various data types, and not just limited in their applications in plasmid genomes.

We agree that the additional background on the credibility of alignment-free methods would strengthen the manuscript. The references suggested by the reviewer have been included in line 64-65:

“Alignment-free genetic distance methods are becoming established tools for the analysis of large genomic datasets, and their usefulness has been validated in both prokaryotes and eukaryotes^{19,23–26}”

2. L87 onward: to which is the 20.64% relative? Total proteins, or total annotated GO BP terms? This should be clarified.

We apologize for the lack of clarity. The pie chart in Figure 1B is the only element which considers the proportion of GO annotations for unique CDSs (duplicates in the dataset were disregarded). This has been stated previously in lines 85-88 and we now also clarified it in Figure 1B legend (line 584). All of the analyses of annotated CDSs (Figure 1B and Supplementary Figure 21 GO bar charts) have been presented relative to the total number of annotated CDSs (i.e. overall frequency of every CDS was taken into account). Previously this was clarified in the figure legends, but we have now also provided details in main text Lines 91-93.

3. L118-130: This section justifying the JI threshold of 0.3 is important. I think Figure 2 may be too technical as a main figure; the authors may consider presenting this figure as a supplementary figure instead. Also, the current Supplementary Figure S3 is quite pixelated – a larger, higher-quality figure would be very helpful.

We firmly believe clear and transparent justification of our approach is an important part of our findings supporting future work on plasmid diversity. Figure 2 summarizes the evidence we considered prior to accepting a 0.3 JI threshold, and provides information on the OSLOM analysis of the original plasmid network. Though we appreciate this is a complex figure, including Figure 2 in the main text encourages the reader to carefully and critically consider our choice of the threshold before the biological relevance of plasmid cliques is addressed. In addition, we have revised some parts of the results which refer to this figure and the 0.3 JI threshold to help interpretability. Finally, we have included a new larger Supplementary Figure 3 with improved quality and a clearer legend.

4. L208: “The gene content of cliques was assessed for all genes occurring five or more times in the dataset.” A brief justification of this approach (i.e. why five?) would be great. If this follows an earlier study, it should be clear. Also, how does this approach cope with gene duplication?

The choice of genes occurring five or more times has now been addressed in lines 242-243: “This threshold was chosen to facilitate computation, and to adequately characterize more prevalent genes.”

In this section of the manuscript (beginning page 8), we explicitly consider the gene content of cliques, therefore gene duplications within plasmid genomes were ignored (i.e. plasmid can only have a gene present or absent). This has now been clarified in the legend of Supplementary Figure 19.

5. L323-327: The use of Jaccard index to assess the similarity of pairwise k-mer profiles is interesting and straightforward. It is simpler than the other methods, e.g. variants of D2 statistics(<https://doi.org/10.1089/cmb.2009.0198> and <https://doi.org/10.1089/cmb.2010.0056>) that incorporate normalization measures e.g. by the probability of specific k-mers found in the dataset. I think the choice of Jaccard index over other k-mer methods should be explained and justified in the text.

We agree this is an important point. We have now addressed it in the Methods section, lines 379-381:

“Also, JI does not weight k-mers based on their abundance, like the popular D_2^* and D_2^S statistics⁴⁴, which would exacerbate the inherent sampling biases towards well-studied species to the dataset.”

6. L352: the parentheses around “-singlet” are not necessary.

Thank you for pointing this out. The parentheses have been removed.

7. L358-359: “after which only cliques were considered with non-complete communities disregarded.” I believe a comma between “considered” and “with” here would make the expression clearer?

We appreciate the reviewer’s attention to detail. A comma has been added.

Thanks for the opportunity to review this paper,
Cheong Xin Chan

Reviewer #2 (Remarks to the Author):

Comments to “Large-scale network analysis captures biological features of bacterial plasmids” by Acman et al.

1. Acman et al. performed network analysis of bacterial plasmids in database based on their shared k-mer content. They successfully classified almost half of the plasmids in database into cliques. They argued that their analyses could provide biological features of plasmids. While the present study is of interest in the research field of plasmid biology, however, this study mainly focused on the methodology of classification. The new findings in the present study are not necessarily clearly shown. If their analyses could classify more plasmids than before, it might be better to show the example(s) of newly classified plasmid(s) more in detail.

Our classification approach assigned 50.21% of plasmids into cliques of size three or more. During the analysis, 48.12% of plasmids were split from the network into singletons and pairs, effectively forming their own disjoint groups and leaving 1.67% of plasmids unclassified. As it is an important aspect of the analysis, we have implemented additions to lines 146 and 154-156 to stress out these points.

Previous studies of plasmid networks (for example: <https://doi.org/10.1186/1471-2105-9-551>, <https://doi.org/10.1093/molbev/msr292>, <https://doi.org/10.3390/pathogens3020356>) considered much smaller datasets and were limited in breadth, covering mainly specific bacterial families or phylum. Moreover, the plasmid networks in these examples were constructed using a BLAST-based approach which estimates genetic distance between plasmids by comparing proportions of known shared genes and protein sequences, and is thus intrinsically biased to well characterised genomes. Most importantly, previous studies have not attempted to classify plasmids into biologically meaningful groups.

As we are using a unique alignment-free approach to studying plasmid genomes, we find it important to establish and benchmark the method. Consequently, a notable segment of the manuscript is focused on introducing the methodology and carefully validating our results. Regardless of the novelty of the approach, we would like to point out the most important biological finding presented throughout the manuscript is the detailed reconstruction of

bacterial plasmid population structure, which recapitulates important biological information such as, gene content, bacterial host, and incompatibility. This allowed studying of HGT events and characterisation of previously undescribed plasmid groups (exemplified by the identification of 105 candidate replicon genes).

In addition to these and in light of your comments, we have included further findings and observations to our manuscript which have been listed below:

- Lines 156-161 and Supplementary Figure 9 – robustness of the genetic signal found on the plasmid backbones
- Lines 177-185 and Supplementary Figure 12 – distinct sub-clades of ‘broad-host-range’ IncP1 plasmids
- Lines 185-190 and Supplementary Figure 13 – potential mistyping of IncY and p0111 plasmids
- Lines 285-290 – HGT in other bacterial taxa is driven by host species’ ecology and life history
- Lines 308-316 – greater importance of transposons in HGT

Lastly, we provide the full genome and meta information on all newly classified plasmids in Supplementary Table 1. Though, the number of cliques with unclassified plasmids is large (Supplementary Figure 14) and extending our analyses would reduce the cohesion of the current manuscript. We believe it would be more appropriate to thoroughly analyse cliques of unclassified plasmids in a follow-up study with dedicated focus.

2. Many plasmids carried accessory genes with transposons. This reviewer could not understand the effects of them on the classification when the same accessory genes or transposons were carried by different plasmids.

We have repeated the analysis with OSLOM community detection algorithm over a range of JI thresholds after having removed 29,913 CDSs associated with transposition, pathogenesis, or resistance. The effects on the classification were insignificant, proving the robustness of the genetic signal of the plasmid backbones. The results are presented in Supplementary Figure 9 and addressed in lines 156-161.

3. They did some comparisons with the classifications based on Rep and MOB typing. Some plasmids even in the same Inc group were not necessarily classified into one clique (e.g. IncP1=IncP and IncP6=IncU), while other plasmids in the same Inc group were into a single clique (e.g. IncL/M., IncQ1=IncP-4). It might be helpful if they explain the reason(s). Related to this, there should be more information for Rep typing, such as IncP-7, IncP-9, and PromA.

To further elaborate the assignment of plasmids into cliques, we have included two case studies of seemingly erratic classifications using standard techniques in the manuscript. In accordance with the reviewer’s suggestion, the first one is that of the IncP1 plasmid type which has been resolved into three cliques. The second concerns an interesting case of IncY and p0111 plasmids which were found both clustered within the same clique and split among two different cliques. The results are presented in Supplementary Figures 12 and 13, and in lines 177-190.

Relating to the reviewer’s comment about additional Rep types, we are aware that the PlasmidFinder database is regularly updated with query sequences for new replicon types. However, at the time this particular analysis was performed the available PlasmidFinder did not include the mentioned plasmid types. The information about the PlasmidFinder database version used has been added in line 361.

4. They did not use plasmid sequences of uncultured (or unidentified) hosts.

The information about bacterial hosts is provided in Supplementary Table 1. In the dataset, 66 plasmids were from unknown bacterial families, 14 were from uncultured bacteria and 37 from *candidatus* species. We admit that the previous version of the manuscript might have given the impression that plasmids from unidentified bacterial hosts had been removed from the analysis, therefore we have emphasized their inclusion in the main text (lines 83-85).

5. Lines 171-173 and Supplementary Table 1. This reviewer could not understand the candidate genes in the table. Were they listed in 'Replicon_candidate'?

Thank you for highlighting this problem. To avoid any confusion, we have now clarified this in the text (lines 205-207). In addition, we made sure to better integrate the information included in the Supplementary Table 1 in the manuscript (lines: 81, 85, 358, 438)

6. Lines 188-189 and Supplementary Figure 3. This reviewer could not understand this part probably because the colour gradient of the nodes in the figure was not clearly shown.

We are sorry the previous figure was difficult to read. We have now included a version of Supplementary Figure 3 at higher resolution and a legend for the colour gradient values.

Reviewer #3 (Remarks to the Author):

This manuscript describes a method of identifying patterns in the similarity network of plasmids via the identification of cliques.

With respect to the network analysis aspect of the work (ignoring the biological relevance of the results), some problems can be identified:

1. The authors propose the use of the OLSOM community detection algorithm to identify cliques, arguing that direct maximal clique identification (max-clique) would be prohibitive computationally, given the NP-complete nature of the problem. However, this is incorrect. Although it's true that max-clique is NP-complete in the worst case, it is fact polynomial in many instances of the problem. In particular, for sparse networks the problem tends to be tractable with the Bron-Kerbosh algorithm, which often runs close to linearly on the number of nodes, and thus usable for networks with up to millions of nodes. For the plasmid network with around 5K nodes, the full enumeration of maximal cliques should take a matter of seconds on a typical laptop. Free implementations of the max-clique algorithm are available in many packages (e.g. in the graph-tool Python library: https://graph-tool.skewed.de/static/doc/topology.html#graph_tool.topology.max_cliques)

It is also important to note that the community detection problem (e.g. via maximization of modularity, or the inference of the stochastic block model) are also NP-Hard in general (but also feasible in many instances), which renders the argument in the manuscript moot. (However, regardless of its feasibility, community detection is not the appropriate ansatz if the objective is to find cliques.)

Thank you for highlighting these useful points and methods. The reviewer's concerns made us realize the presentation of the OSLOM-based approach and JI threshold were previously not clear enough. We have retained our original method for detecting cliques, but now provided extensive clarification and justification for the method as well as the importance of edge weights (lines 99-143 and Supplementary Figure 8).

We had considered employing a max-clique algorithm to classify the network as it is fast, exact and well-established. However, this would have required us to consider the JI threshold as a rigid boundary which transforms between-plasmid genetic distances (weighted edges) into binary values. As plasmid genomes are highly plastic, can undergo homologous recombination and exchange genetic elements, the distribution of pairwise genetic distances can vary greatly within and across different plasmid groups. Therefore, we considered it important to employ pairwise genetic distances in the analysis.

To highlight the importance of considering genetic distances when classifying plasmid genomes, we have implemented the approach suggested by the reviewer across a range of JI thresholds using the graph-tool Python library. However, we found that the identification of maximal cliques was not satisfactory, relative to our original community detection method. For example, the number of identified cliques of size three or more is far larger compared to our original method and grows exponentially at lower threshold values, which reflects a large amount of shared genetic elements scattered across plasmid genomes. Likewise, we obtain a far higher number of plasmids being assigned to multiple cliques and lower values of NMI when compared to replicon-based classification of plasmids.

2. The similarity network is converted into a binary network via thresholding. Although the authors discuss briefly the chosen threshold value, its choice is likely to have a very strong impact on the existence of cliques in the resulting network. Therefore, it is absolutely necessary to evaluate the robustness of the results on the choice of the threshold value.

At no point do we convert our similarity network into a binary one. Though, we agree that the choice of a threshold value can be critical and we tested this carefully, as shown in Figure 2 and Supplementary Figure 6. The threshold was introduced to optimize the performance of the OSLOM algorithm, and, as shown in the figures, it has limited impact on our results around the chosen value of 0.3 (+/- 0.025). The robustness of our results has been additionally addressed in lines 156-161 and Supplementary Figure 9.

The above points are serious problems with the current analysis, and it is imperative they are addressed before the the manuscript is further considered for publication.

We hope we satisfactorily addressed the concerns raised by reviewer 3.

REVIEWERS' COMMENTS:

Reviewer #2 (Remarks to the Author):

Comments to "Large-scale network analysis captures biological features of bacterial plasmids" by Acman et al.

The revised manuscript by Acman et al. rather improved according to the reviewer's comments and questions. This reviewer has only one comment.

Lines 177-190 and 315. Because the authors used plasmid group of E. coli nomenclature, 'IncP1' might be better to be shown with 'IncP' (i.e. 'IncP/IncP1' or just 'IncP'). These two groups are identical, but they have been differently classified in Pseudomonas or Escherichia.

Reviewer #3 (Remarks to the Author):

I'm satisfied with the authors' responses.

Response to Reviewers

A point-by-point response to all reviewers' comments is included below.

REVIEWERS' COMMENTS:

Reviewer #2 (Remarks to the Author):

The revised manuscript by Acman et al. rather improved according to the reviewer's comments and questions. This reviewer has only one comment. Lines 177-190 and 315. Because the authors used plasmid group of E. coli nomenclature, 'IncP1' might be better to be shown with 'IncP' (i.e. 'IncP/IncP1' or just 'IncP'). These two groups are identical, but they have been differently classified in Pseudomonas or Escherichia.

We are pleased to hear extensive improvements made in our revised manuscript have been recognized and would like to thank you for contributing with your suggestions and comments. Also, thank you for noticing the oversight in plasmid nomenclature. The change has been made in the main text, but also in Supplementary Figure 12.

Reviewer #3 (Remarks to the Author):

I'm satisfied with the authors' responses.

We are happy to hear our responses have been satisfactory. Thank you for your valuable input which helped increase the credibility of this study